# Class I histone deacetylases inhibition reverses memory impairment induced by acute stress in mice

Heidy Martínez-Pacheco[1], Rossana Citlali Zepeda[2], Ofir Picazo[3], Gina L. Quirarte[1]*, Gabriel Roldán-Roldán[4]*

1 Departamento de Neurobiología Conductual y Cognitiva, Instituto de Neurobiología, Universidad Nacional Autónoma de México, Campus Juriquilla, Juriquilla, Querétaro, México, 2 Centro de Investigaciones Biomédicas, Universidad Veracruzana, Xalapa, Veracruz, México, 3 Escuela Superior de Medicina, Instituto Politécnico Nacional, Ciudad de México, México, 4 Laboratorio de Neurobiología Conductual, Departamento de Fisiología, Facultad de Medicina, Universidad Nacional Autónoma de México, Ciudad de México, México

* ginaqui@unam.mx (GLQ); gabaergico@gmail.com (GR-R)

## Abstract

While chronic stress induces learning and memory impairments, acute stress may facilitate or prevent memory consolidation depending on whether it occurs during the learning event or before it, respectively. On the other hand, it has been shown that histone acetylation regulates long-term memory formation. This study aimed to evaluate the effect of two inhibitors of class I histone deacetylases (HDACs), 4-phenylbutyrate (PB) and IN14 (100 mg/kg/day, ip for 2 days), on memory performance in mice exposed to a single 15-min forced swimming stress session. Plasma corticosterone levels were determined 30 minutes after acute swim stress in one group of mice. In another experimental series, independent groups of mice were trained in one of three different memory tasks: Object recognition test, Elevated T maze, and Buried food location test. Subsequently, the hippocampi were removed to perform ELISA assays for histone deacetylase 2 (HDAC2) expression. Acute stress induced an increase in plasma corticosterone levels, as well as hippocampal HDAC2 content, along with an impaired performance in memory tests. Moreover, PB and IN14 treatment prevented memory loss in stressed mice. These findings suggest that HDAC2 is involved in acute stress-induced cognitive impairment. None of the drugs improved memory in non-stressed animals, indicating that HDACs inhibitors are not cognitive boosters, but rather potentially useful drugs for mitigating memory deficits.

## Introduction

The effects of stress on learning and new memories formation have been extensively studied, yielding highly variable results, facilitating, or impairing memory depending on the type of stress, intensity, duration, learning task, experimental paradigm, and the moment in which stress occurs (inside or outside the learning context) [1–7]. Multiple studies in animals and humans have shown that the effects of acute stress on memory are critically dependent on

**Data Availability Statement:** All relevant data are within the manuscript and its Supporting Information files.

**Funding:** This research was funded by the grant DGAPA-PAPIIT-UNAM (IN209822), Postdoctoral fellowship from DGAPA-UNAM (2020-2022) to HM-P, COFAA, and SIP-IPN. The funders had no role in study design, data collection and analysis, decision to publish, or preparation of the manuscript.

**Competing interests:** The authors declare that they have no known competing financial interests or personal relationships that could have appeared to influence the work reported in this paper.

their temporal relationship [7–12], where it is assumed that a stressful event improves memory if it is experienced within the context of the learning episode and, on the contrary, negatively affects memory when it is experienced outside the learning context [13]. Stress and/or release of stress hormones may lead to the deterioration of cognition through various mechanisms, including epigenetic regulation [1, 14–17]. Many studies have focused on the role of the hippocampus in this phenomenon, providing evidence that this brain structure is crucial for glucocorticoids (GCs)-dependent memory consolidation [18].

Histone deacetylation, mediated by HDACs enzymes, is a typical epigenetic regulatory mechanism involved in learning, memory formation, and stress [15, 19–21]. In this regard, neuron-specific over-expression of HDAC2 impaired memory formation in adult mice, whereas HDAC2 deficiency resulted in memory facilitation, like that induced by nonselective HDACs inhibitors treatment [1, 22]. In addition, high-intensity stress reduces histone acetylation [23, 24] and impairs recognition memory through regulation of GCs receptors activity and HDAC2 expression.

On the other hand, elevated HDAC2 levels are involved in the cognitive decline associated with neurodegeneration in CK-p25 mice, through mechanisms involving glucocorticoid receptor 1 (GR1) phosphorylation and interaction with the HDAC2 promoter-glucocorticoid response elements (GRE), whereas the prevention of HDAC2 upregulation rescues memory capacities in a model Alzheimer's disease [25].

Given this evidence, it is reasonable to assume that acute stress-induced memory impairment may be mediated, at least in part, by HDAC2 activity, a relationship that remains unclear. Pharmacological inhibition of these enzymes prevents acute and chronic stress-induced anxiety [26], as well as cognitive deficits observed in neurodegenerative diseases [27, 28]. The present study aimed at investigating whether acute extrinsic stress caused by forced swimming 30 min before learning affects short- and long-term memory using three different hippocampus-dependent memory tasks. We also determined whether acute swimming stress induces an increase in plasma levels of corticosterone and HDACs. Finally, we tested the effect of two HDACs inhibitors, 4-phenylbutyrate (PB) and IN14, on acute stress-induced memory blockade. We chose these two drugs for comparative purposes, based on previous studies showing that PB is a reversible inhibitor of classes I and II HDACs with well-known capacity to reverse memory deficiencies, while IN14 appears to be more selective for class I but with similar biochemical and behavioral efficacy [29, 30]. Our findings suggest that HDAC2 is involved in extrinsic stress-induced cognitive impairment.

## Materials and methods

This study was carried out in accordance with the protocols approved by the Committee on the Use of Live Animals in Teaching and Research of the UNAM (FM/DI/036/2017), which comply with the "International Guiding Principles for Biomedical Research Involving Animals" [31]. Efforts were taken to minimize the animals' suffering throughout the experiments.

### Animals

Seventy-six three-month-old (22 to 25 g) male CD1 mice were obtained from the colony of the Faculty of Medicine, Universidad Nacional Autónoma de México (UNAM). They were housed in groups of 8 in 34.5 X 49 X 17 cm polycarbonate boxes with sterilized pine shavings and compressed cardboard tubes of different sizes for the mice to hide in and make their burrows, in a temperature-controlled room (22 ± 1°C) with a 12 h light-dark cycle (lights on at 07:00 A. M.) and *ad libitum* access to food (standard rodent chow) and water for at least one week

before the beginning of the experiments. All animals were habituated to handling once a day for three days prior to any procedure.

## Chemicals and reagents

The compound IN14 was synthesized and characterized by our group [29, 30]. 4-phenylbuty-rate (PB), Pentobarbital, potassium dichromate (K2Cr2O7), and MS-grade ammonium formate were purchased from Sigma, St Louis, MO, USA. MS-grade methanol and formic acid were purchased from Merck, S. A de C.V. (México). Deionized water (resistivity 18.2 MΩ-cm) for sample pre-processing and mobile phase preparation was obtained from a water purification system (ThermoFisher Scientific, México).

## Groups and treatments

Independent groups of mice were randomly divided and subjected to different experimental manipulations as described in Fig 1. The variables were to induce or not stress in mice by forced swimming, and the systemic (i.p.) treatment with HDACs inhibitors: PB (100 mg/kg/day), IN14 (100 mg/kg/day) or vehicle (physiological saline), for 2 days (Day 1 and Day 2) at 9:00 a.m. On Day 2, behavioral experiments began.

## Acute swim stress

The forced swimming method employed in this study was based on the original procedure [32] and was used to induce acute stress. Briefly, mice were placed individually in a cylindrical

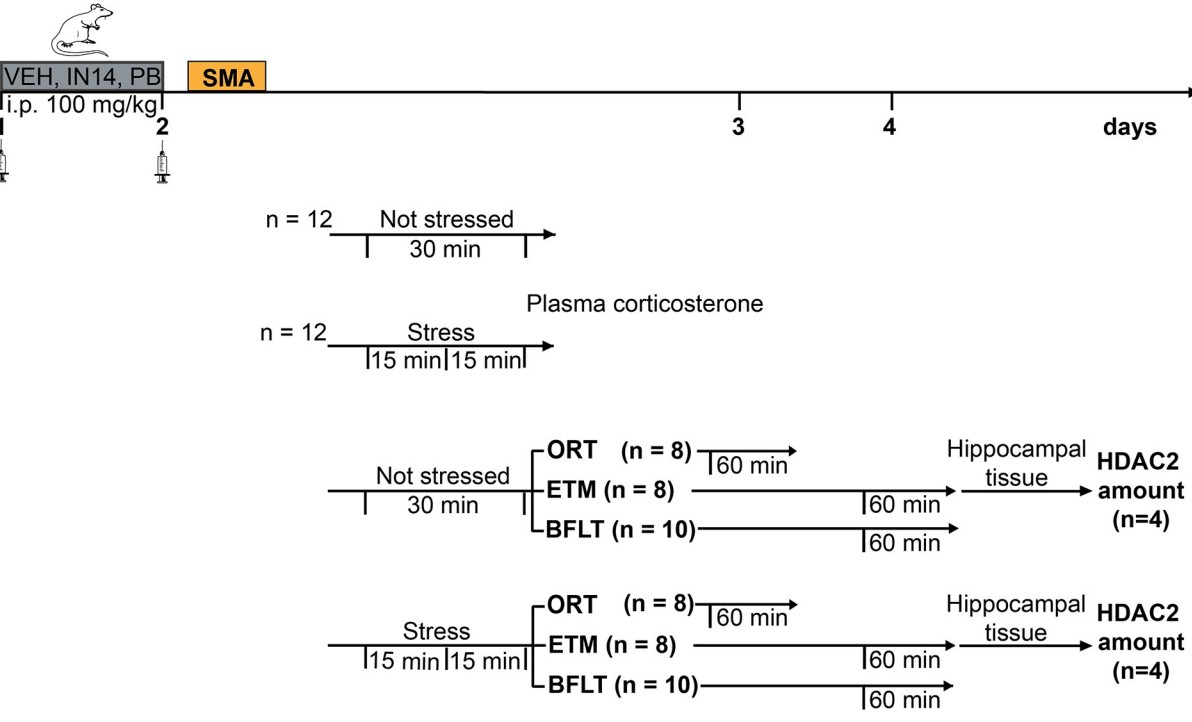

**Fig 1. Experimental design.** All mice were habituated to the experimental room for 5 minutes one day before training on Day 1. On Day 2, spontaneous motor activity (SMA) was assessed in all groups. Ten min later, one group was stressed through forced swimming for 15 min and the other remained in their home cage. Fifteen minutes later stressed and non-stressed animals were sacrificed and blood samples were collected for plasma corticosterone levels analysis. In another experimental series, independent groups of mice were trained in the Object recognition test (ORT), Elevated T maze (ETM), and Buried food location test (BFLT). Long-term memory (LTM)-tests were carried out 24 h later (Day 3) for ORT groups and 48 h later (Day 4) for ETM and BFLT groups. Sixty minutes later, hippocampal tissue samples were collected for ELISA assays to determine HDAC2 content in the ETM groups only.

glass tank (25 cm height × 19 cm diameter) filled with water (25 ± 1˚C) to a depth of 15 cm for 15 min. Immediately after the 15-minute swim, mice were removed from the tank, dried with a towel, and put in a warming cage (37˚C) that contained a heating pad covered with towels for 15 additional min, i.e., forced swimming began 30 minutes before training. After training and/or short-term memory (STM) testing, mice were returned to their home cage. Forced swimming occurred between 10:00 and 12:00 h.

## Plasma corticosterone levels

Thirty minutes after forced swimming, mice were carefully handled and euthanized by decapitation and trunk blood was collected in heparinized (500 IU ml–1) tubes. Samples were centrifuged at 12,000 rpm for 20 min at 4˚C, plasma was collected and stored at -20˚C. Plasma corticosterone concentration was measured using the corticosterone enzyme-linked immunosorbent assay kit (ab108821, Abcam), following the manufacturer's instructions. The absorbance at 450 nm was immediately read on a microplate reader (Biorad-Ultramark). Each sample was run in triplicate and the mean optical density was converted to concentration.

## Spontaneous motor activity (SMA)

Motor activity is a suitable index to assess the effects of pharmacological agents in addition to rule out any motivational or motor alteration that may affect the performance of animals in behavioral tests. The Opto-Varimex activity-monitoring system was used in which each mouse was placed for 5 minutes in an acrylic box (51.1 X 9.5 X 69.2 cm) provided with a sensors system that allows the total, ambulatory and vertical activity of the animal to be counted automatically. The SMA was measured 10 min before swim stress. Results are presented as the mean ± SEM of the number of counts in 5 min of ambulatory, vertical and total activity (which also includes mouse movements when not scrolling).

## Behavioral procedures

All behavioral experiments were carried out between 10:00 and 14:00 h and the sessions were recorded using a video camera. On every test day, mice were habituated for 1 h before testing by placing their cage in the experimental room with no water bottle or feeder bin for 5 min.

**Object recognition test (ORT).** Recognition memory was assessed using the ORT [33]. This non-rewarded learning task is based on the spontaneous exploratory behavior of rodents. The experiments were conducted in an open acrylic arena of 30 × 23 × 21,5 cm. The experimental protocol consists of habituation, acquisition, and test phases. Before starting the experiment, animals were habituated to handling once a day for three days. Then, mice were habituated to the open field for 5 min without any object and submitted to the acquisition trial 24 h later. During acquisition, mice were placed in the experimental arena with two identical objects (50 ml beakers, 4.2 cm diameter, 5.6 cm height; A-A') located 10 cm apart, equidistant from the walls, and were allowed to freely explore them for 5 min. Short-term memory was tested 60 min later in the same way as the acquisition session except that one of the identical objects (A') was replaced by a novel object (B; a blue Lego block, 2.5 × 2.5 × 5 cm). The long-term memory test was carried out 24 h after acquisition in an identical way, but object B was replaced by another novel object (C; a LED Bulb of 55 mm). The three objects used in this study were selected after confirmation that mice preferred them equally through a pilot study in which the objects were presented in pairs to independent groups of mice only once as follows: A and A', A and B, and A and C, showing that they had no preference for any one in particular. Each animal had equal opportunities to explore both objects all around, ruling out the possibility of side bias. To eliminate olfactory cues, the testing box and the objects were cleaned

with a 10% ethanol solution after each trial. Exploration was defined by distinctive behaviors such as turning the head towards the object and sniffing it at less than 1cm or touching the object with the nose and forepaws. The exploration time of each object was recorded and the discrimination index (DI) was calculated as follows:

Discrimination index = (TN-TF)/(TN+TF) where TN = time spent with the novel object and TF = time spent with the familiar object. A positive score indicates a preference for the novel object and a negative score for the familiar object [34, 35].

**Elevated T maze (ETM).** The ETM apparatus was made of acrylic, had three arms of equal dimensions (33 cm × 5 cm), and was elevated 50 cm above the floor. One arm, enclosed by walls 25 cm high, was perpendicular to the two opposed open arms. Independent groups of control and treated animals (n = 8/ group) were trained. During the acquisition session, each mouse was placed at the end of the enclosed arm of the maze and the time to exit this arm and enter with all four paws in any of the open arms was recorded (acquisition latency, AL). The same maneuver was then repeated in subsequent trials at intervals of 1 min (acquisition latencies AL1, AL2, AL3, and AL4). The permanence in the enclosed arm for 180 seconds was considered the learning criterion, thus if a mouse did not leave the arm during this period, the trial was ended and a score of 180 was assigned. Being on an open arm is an aversive experience since rodents have an innate fear of height and openness [36, 37]. Thus, when the animal is repeatedly placed inside the enclosed arm and allowed to explore the maze, they acquire an inhibitory avoidance of the open arms. To verify that animals did not have any motivational or motor impediment that prevented them from performing the task, once each mouse had reached the learning criterion it was placed at the end of one open arm, and the latency to leave this arm and enter with all four paws to the enclosed arm was recorded (escape latency, EL). Forty-eight hours later long-term memory was evaluated in a single retention trial identical to the acquisition session where the time to leave the enclosed arm was recorded (retention latency, RL).

**Buried food location test (BFLT).** The BFLT was first described in 1971 [38]. This test has been used to evaluate olfactory acuity under physiological and pathological conditions [39]. A modification of the BFLT was developed by our group in order to assess spatial learning and memory [40]. It comprises two sessions, the training session, and the test (Fig 2). Before the training session, animals were fasted for 24 h with *ad libitum* access to water. The

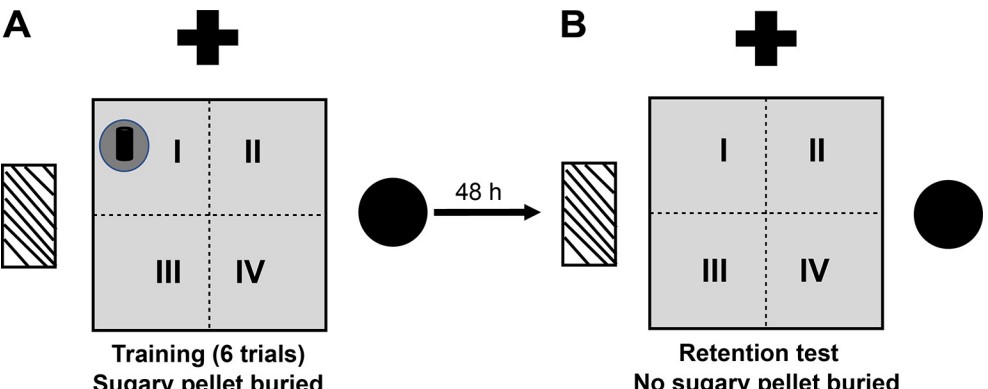

**A**  Training (6 trials)
Sugary pellet buried

**B**  Retention test
No sugary pellet buried

48 h

**Fig 2. Diagram of the buried food location test.** Top view of the arena, where the buried food reward (fruit loop) location in quadrant I (black dot) is indicated. (A) Acquisition session consisting of training 6 trials, where the mice were placed randomly inside the box. (B) Long-term spatial memory retention testing without food reward. Plus symbol, rectangle with diagonal stripes and circle correspond to the real spatial clues that were placed on the walls of the experimental room.

training session (acquisition) consisted of 6 trials (1 min inter-trial interval) in which mice were placed randomly in an acrylic box (40 x 52 x 20 cm) covered with a sawdust layer of 3 cm and a highly palatable food (sugary pellet, "Kellog's Fruit Loops" to which they were previously familiarized to avoid neophobic responses) was buried 1 cm under the sawdust in a fixed quadrant of the box. The location of the pellet was the same in all the trials. The walls of the box exhibited spatial clues (colored geometric figures, placed at a height of 25 cm). Mice were allowed to freely explore the testing box until finding the pellet (acquisition latency) and permitted to eat it for 5 s. If mice did not find the pellet within 60 s of the first trial, they were gently guided to it. After each trial, mice were returned to their home cage, the testing box was cleaned with a 10% ethanol solution, and the sawdust was removed to eliminate odorous marking. Animals were habituated to the testing box for 5 min twice, 24 and 1 h before the training session, to reduce novelty-induced exploratory activity. Forty- eight h after training, long-term memory was evaluated in a single retention test where the food was removed from the sawdust and mice were allowed to freely explore the box for 60 s. The time to reach the precise location where the pellet was buried (same location as during training) and the time spent exploring the pellet location quadrant were measured. All sessions were video recorded and analyzed offline by an experimenter blind to the treatments. Results are expressed as the time to reach the target (retention latency) and the time spent searching for the pellet in the target quadrant.

## Hippocampal HDAC2 activity

**Isolation of nuclear proteins.**   A representative sample of mice from the EPM group was used to perform the HDAC2 expression assay to verify its stress-induced increase as previously reported (55, 70), and to evaluate the effect of PB and IN14. Thus, after completion of behavioral testing, mice were sacrificed by decapitation and the hippocampi were obtained. Nuclear proteins were isolated from 100 mg mice hippocampal tissue fragments using 1 mL of cold buffer A (10 mM Hepes pH 7.5; 10 mM KCl; 1 mM EDTA; 1 mM EGTA, 1 mM DTT; 1 mM PMSF; 1 μg/mL Aprotinin; 1 μg/mL; and Leupeptin). It was incubated for 15 min on ice, then 25 μL of 10% NP40 solution was added. Finally, the samples were centrifuged for 1 min at 10,000 rpm at 4° C, and the supernatants were stored at -70° C (cytoplasmic extract) until use. The pellets (nuclear extracts) were resuspended in 250 μL of Buffer C (20mM Hepes pH 7.9; 400mM NaCl; 1 mM EDTA; 1 mM EGTA, 1 mM DTT; 1 mM PMSF; 1 μg / mL Aprotinin; and 1 μg / mL Leupeptin). The tubes were vortexed for 30 min at 4°C. It was centrifuged for 5 min at 9000 rpm at 4°C and the supernatant (nuclear extract) was collected, aliquoted, and stored at -70°C until use. The Bradford method was used to calculate the concentration of nuclear proteins.

**EpiQuik HDAC2 Assay Kit (Colorimetric).**   HDAC activity was determined using a Fluor de Lys HDAC2 Activity Colorimetric Kit, following the manufacturer's instructions. This kit semi-quantitatively measures the amount of HDAC2. The absorbance signal was detected at 450 nm using a BioTek Epoch Microplate Spectrophotometer. In this experiment, the HDAC inhibitor PB was used as a positive control.

## Data analysis

All data are expressed as the mean ± standard error of the mean (SEM) and were analyzed using ANOVAs followed by post hoc Sidák or Tukey tests when appropriate. GraphPad Prism® software version 6.01 was used to perform all analyses. Differences between groups were considered statistically significant if $p < 0.05$.

## Results

### Effect of PB and IN14 on plasma corticosterone level induced by acute swim stress

Forced swimming stress for 15 min induced a ~3-fold increase in plasma corticosterone level compared to the control non-stressed group, as depicted in Fig 3. This figure also shows the effect of IN14 and BP pre-treatment on the same parameter. The two-factor ANOVA showed significant differences for stress but not for drug treatment ($F_{(1, 18)}$ = 72.42, p <0.0001; $F_{(2, 18)}$ = 3.264, p = 0.0617, respectively), with a significant interaction between the two factors ($F_{(2, 18)}$ = 6.334, p = 0.0083). Post-hoc Tukey test revealed that PB and IN14 significantly prevented stress-induced plasma corticosterone rise (VEH-S vs. PB-S and IN14-S, p < 0.01), unlike the VEH-S group (VEH vs. VEH-S, p < 0.001).

### Effects of PB and IN14 on spontaneous motor activity of mice

The total counts of spontaneous motor activity 90 min after each of the HDACs inhibitors administration is presented in Fig 4. One-way ANOVA revealed that treatment with IN14 (100 mg/kg, i.p.) or PB (100 mg/kg, i.p.) did not significantly alter the total and ambulatory spontaneous motor activity, (total: $F_{(3, 28)}$ = 2.561, p = 0.0749; ambulatory: $F_{(3, 28)}$ = 2.880, p = 0.0536). However, One-way ANOVA ($F_{(3, 28)}$ = 10.12, p = 0.01) followed by post-hoc analysis showed that treatment with IN14 or PB significantly increased vertical activity (VEH vs. IN14, p< 0.01; VEH vs. PB p< 0.01).

### PB and IN14 reversed acute stress-induced hippocampus-dependent memory impairment

**Novel object recognition test (NORT).** Acute swim stress-induced episodic memory impairment was assessed by using the NORT (Fig 5). Discrimination index, an indicator of

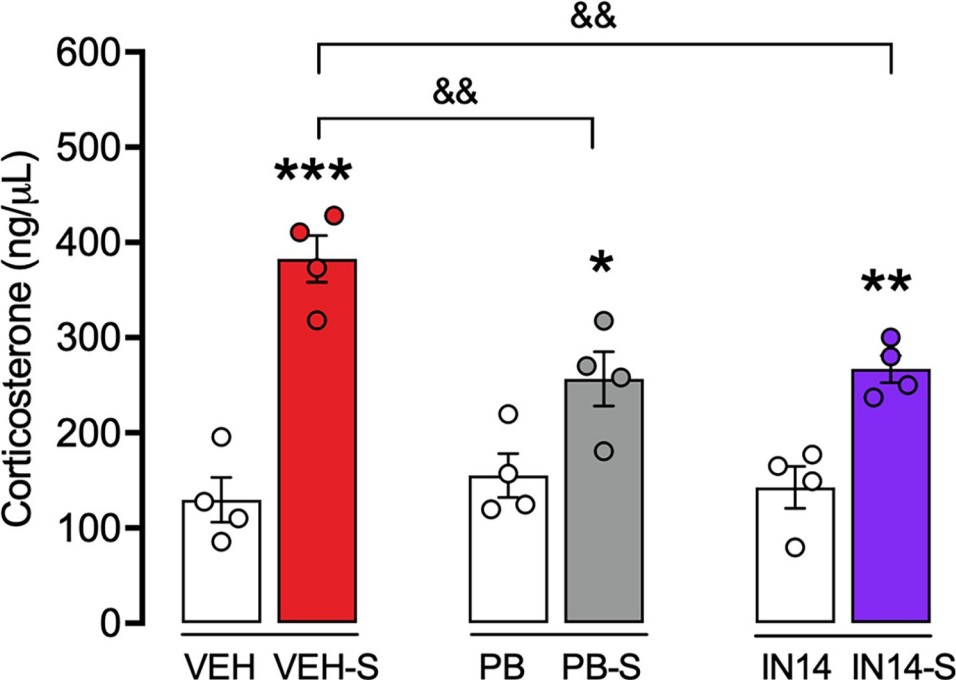

**Fig 3. Effect of IN14 and PB on acute swim stress-induced increase in plasma corticosterone level.** All values are expressed as mean ± SEM (n = 4) * P < 0.05; ** P < 0.01; *** P < 0.001; && P < 0.01, Tukey test. VEH, PB, and IN14 mice were not subjected to any behavioral manipulations. VEH: vehicle; PB: 4-phenylbutyrate.

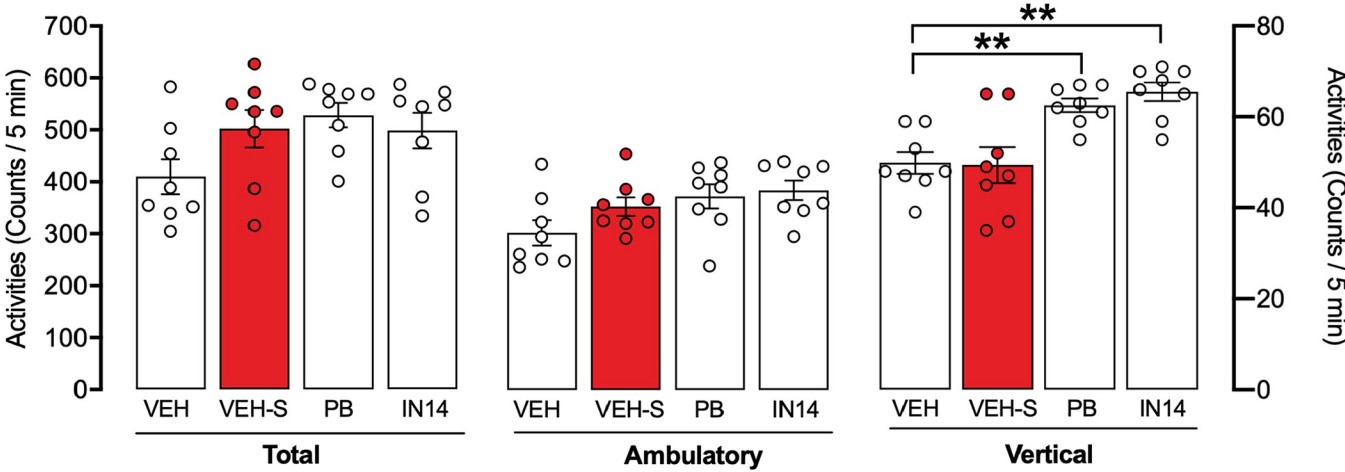

**Fig 4. Motor activity after PB and IN14 injection in mice exposed to acute swim stress.** The results are expressed as the mean ± SEM of spontaneous motor activity: total, ambulatory, and vertical, (n = 8 per group). Vertical activity (right scale): ** $p < 0.01$, Tukey test. VEH: vehicle; PB: 4-phenylbutyrate.

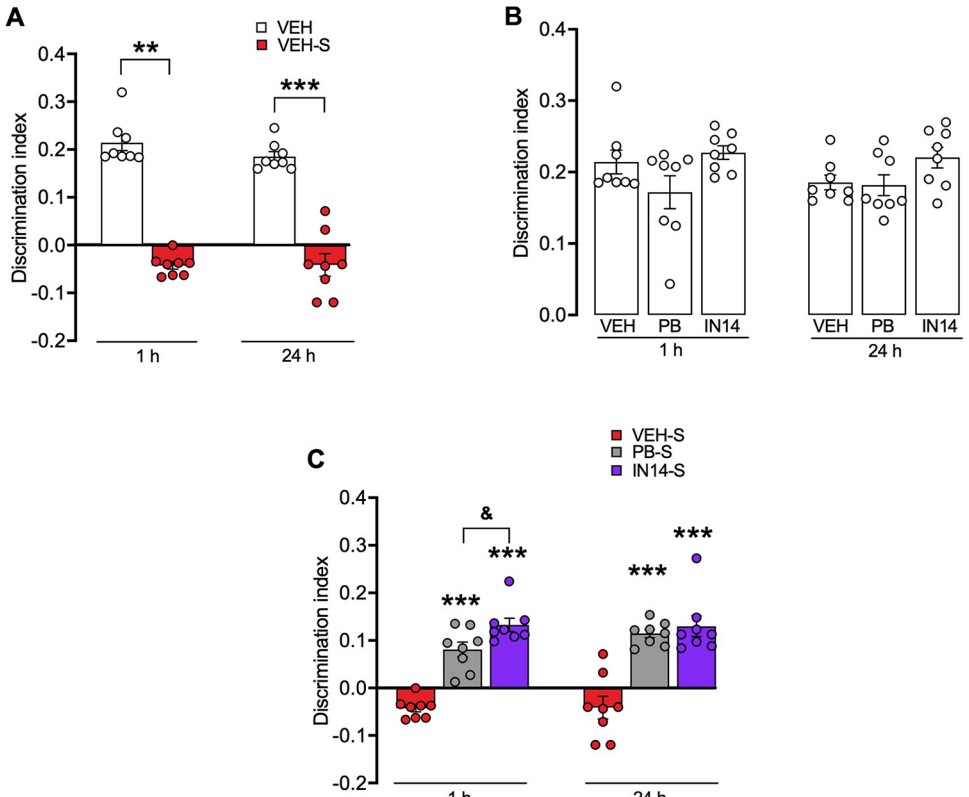

**Fig 5. Effect of PB and IN14 on swim stress-induced short- and long-term memory impairment in the NORT.** A) Swim stress impaired short- and long-term memory retrieval. B) Lack of effects of PB and IN14 (both 100 mg/kg) in non-stressed mice performance. C) Effects of PB and IN14 on swim stress-induced short- and long-term memory impairment. ** $p < 0.01$; *** $p < 0.001$ vs. is respective vehicle group. $^{\&}p < 0.05$, PB vs. IN14. Data are expressed as mean ± SEM, n = 8 per group. VEH: vehicle; PB: 4-phenylbutyrate.

recognition memory, was found significantly reduced in stress-exposed mice 30 min after swimming compared to the vehicle group (one-sample t-test: t = 13.96, p <0.0001) and 24 h (one-sample t-test: t = 8.788, p <0.0001, Fig 5A). HDACs inhibitors treatment before training did not improve short nor long-term memory in non-stressed mice (One-way ANOVA: $F_{(2, 21)}$ = 2.793, p = 0.0840; $F_{(2, 21)}$ = 2.6, p = 0.0977 respectively; Fig 5B), but reverted short and long-term memory deficit observed in swim-stressed mice (One-way ANOVA: $F_{(2, 21)}$ = 48.13, p<0.0001 and $F_{(2, 21)}$ = 24.41, p<0.0001, respectively). Post-hoc test confirmed that animals submitted to swim stress showed significantly lower discrimination indexes in the NORT when compared to the vehicle group (p < 0.0001), indicating that swim stress impairs short and long-term recognition memory. This impairment was rescued by systemic administration of either one of the HDACs inhibitors (Fig 5C). Interestingly, IN14-treated mice exhibited a significantly higher discrimination index than the PB-treated group (PB-S vs. IN14-S, p = 0.026), suggesting that it was more efficient in rescuing the stress-induced memory impairment. Finally, vehicle, PB, and IN14 groups did not differ in the total exploration time of the two objects during either training or testing.

**Elevated T maze (ETM).** Fig 6 illustrates the effects of PB and IN14 on the acquisition, escape, and retention latencies in the ETM. Two-way mixed ANOVA revealed a significant effect for trial ($F_{(3, 126)}$ = 666.8, P<0.0001) and drug treatment ($F_{(5, 42)}$ = 24.87, p<0.0001) as well as a trial x treatment interaction ($F_{(15, 126)}$ = 16.76, p<0.0001)). The post hoc Tukey test revealed that swim stress slowed down acquisition in the ETM reaching significant differences in the AL2 (VEH-S vs. VEH, PB-S, IN14-S, p<0.0001; Fig 6B). In all the groups, latency to enter the open arms with all four paws increased gradually in the last two training trials, suggesting the acquisition of inhibitory avoidance towards the open arms (AL1 vs. AL3, AL4, p <0.0001, Sidák test), reaching the learning criterion by remaining 180 s in the enclosed arm. One-way ANOVA revealed that neither stress nor treatment with IN14 and PB modified the latencies to escape from the open arm (EL: $F_{(5, 42)}$ = 9.338, p = 0.0963; Fig 6C). However, one-way ANOVA ($F_{(5, 42)}$ = 25.41, p = 0.0001) followed by the Tukey test showed that swim stress impaired long-term memory retention, an effect that was reversed by systemic administration either HDACs inhibitor (p<0.0001 for VEH vs. VEH-S, VEH-S vs. PB-S and VEH-S vs. IN14-S; Fig 6B).

**Buried food location test (BFLT).** Fig 7 demonstrates the effects of PB and IN14 on the acquisition and retention of the BFLT. Two-way mixed ANOVA showed a significant effect of trial ($F_{(5, 270)}$ = 191.6, p <0.0001) and treatment ($F_{(5, 54)}$ = 5.530, p = 0.0003), as well as a trial x treatment interaction ($F_{(25, 270)}$ = 1.626, p = 0.0332). Post hoc Sidák test showed that there was a significant reduction in the acquisition latencies (AL1 vs. AL3, AL4, AL5, AL6, ***p < 0.001) in all groups, indicating that mice were able to learn the location of the food reward regardless of treatment. However, stressed animals performed poorly in the last training trial (p< 0.001, VEH vs. VEH-S for AL6, Tukey test), although, no significant differences between the stressed groups treated with VEH, PB, and IN14 (VEH-S vs. PB-S, p = 0.9830, and VEH-S vs. IN14-S, p = 0.6990, Fig 7A) were found.

Long-term memory for the location of the buried food was tested 48 h later using a 60-s free-search probe trial. One-way ANOVA ($F_{(5, 54)}$ = 30.62, p = 0.0001) followed by the Tukey test, showed that swim stress impaired long-term memory retention evidenced by a significant increase in latency to reach the food reward location (VEH vs. VEH-S, P<0.001; Fig 7B), an effect that was reversed by IN14 (VEH-S vs. IN14-S, p = 0.0291), but not by PB (VEH-S vs. PB-S, ns = 0.0885). Fig 7C. shows the time mice spent in the target quadrant searching for the food reward during the 60 s probe trial. One-way ANOVA revealed that there were significant differences between the groups ($F_{(5, 54)}$ = 21.86, p< 0.001). The post-hoc test showed that there were significant differences in time exploring the target quadrant between VEH vs.

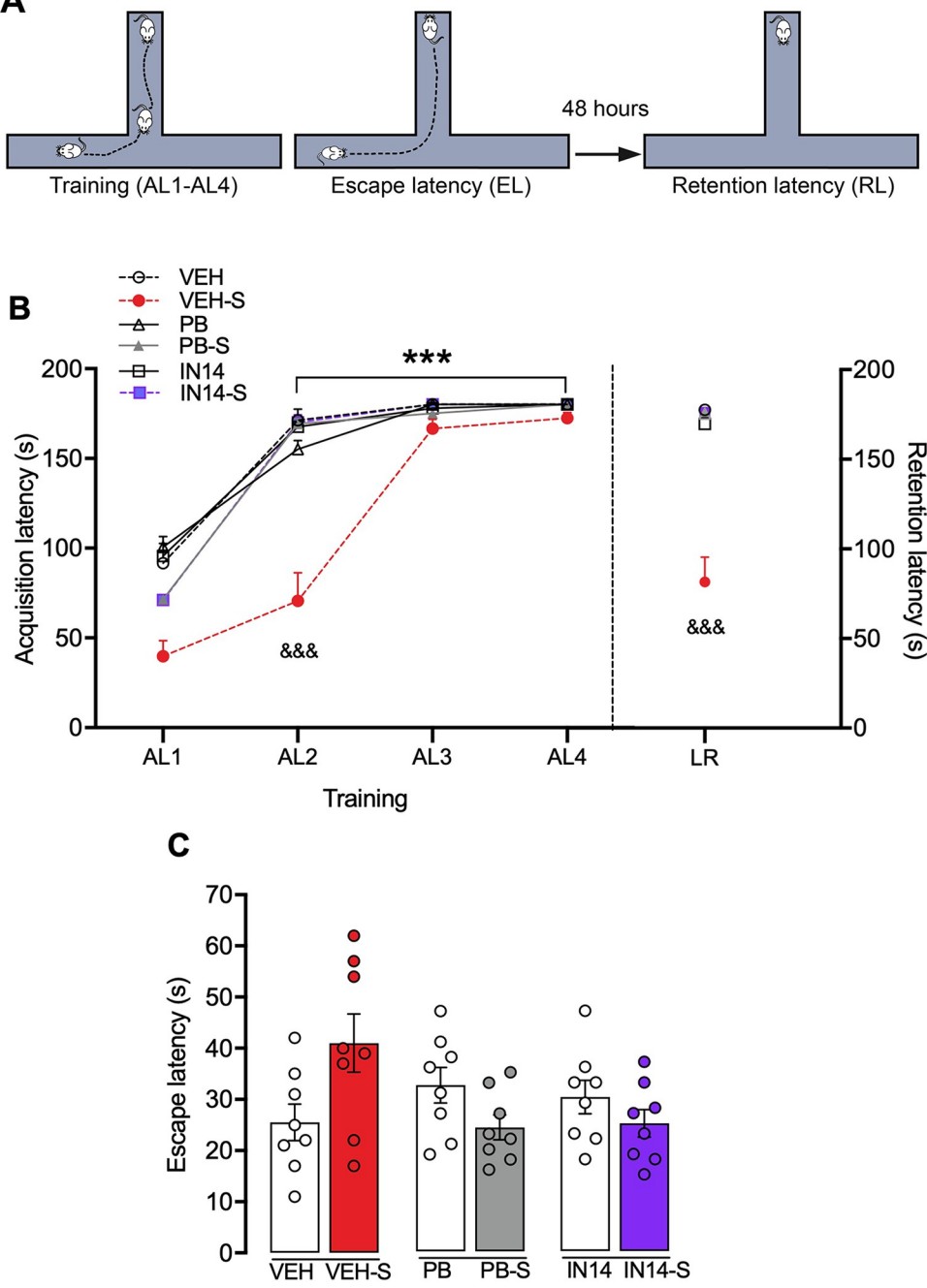

**Fig 6. Effect of PB and IN14 on swim stress-induced long-term memory deficit in the ETM.** A) Experimental protocol of the ETM task (see 2.4, Methods section). B) ETM learning curves during the training session. AL1 vs. AL2, AL3, and AL4, ***p <0.001, Sidák test for all groups, except AL2 of the VEH-S group; VEH-S vs. VEH, PB-S, IN14-S, &&&p<0.001 for AL2, Tukey test. Retention latencies are shown on the right axis. VEH vs. VEH-S; VEH-S vs. PB-S and VEH-S vs. IN14-S, &&&p< 0.001, Tukey test. C) Escape latencies (EL), no significant differences were observed among the groups, p = 0.0963, one-way ANOVA. Data expressed as mean ± SEM, n = 8 per group. AL: acquisition latency; VEH: vehicle; PB: 4-phenylbutyrate.

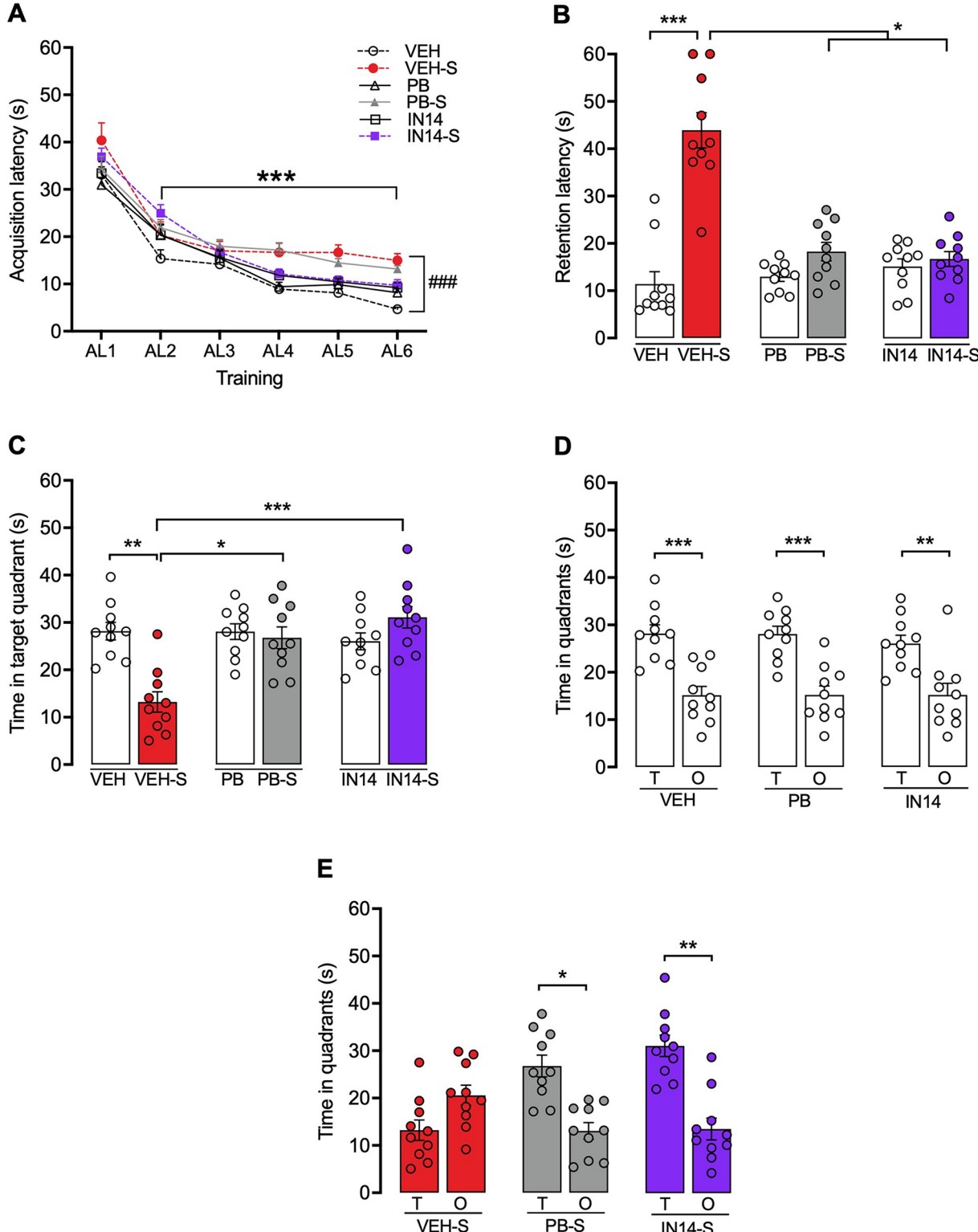

**Fig 7. Effect of PB and IN14 on swim stress-induced memory impairment in the BFLT.** A) Learning curves of mice during the training session, *** $p < 0.001$ for AL1 vs. AL3, AL4, AL5, AL6, Sidák test; ### $p < 0.001$, VEH vs. VEH-S, Tukey test. B) Retention latencies (LR) in the long-term retrieval test. C) Time in the target quadrant during long-term retrieval testing. D) and E) Time spent in the target (T) and opposite O) quadrants of the arena during long-term retrieval testing. The VEH group and the groups treated with PB (-/+ stress) and IN14 (-/+ stress) spend more time in the target quadrant than in the opposite one, unlike the VEH-S. Data are presented as mean + S.E.M. n = 10. AL: acquisition latency; VEH: vehicle; PB: 4-phenylbutyrate. * $p < 0.05$, **$p < 0.01$, ***$p < 0.001$, post hoc Tukey test.

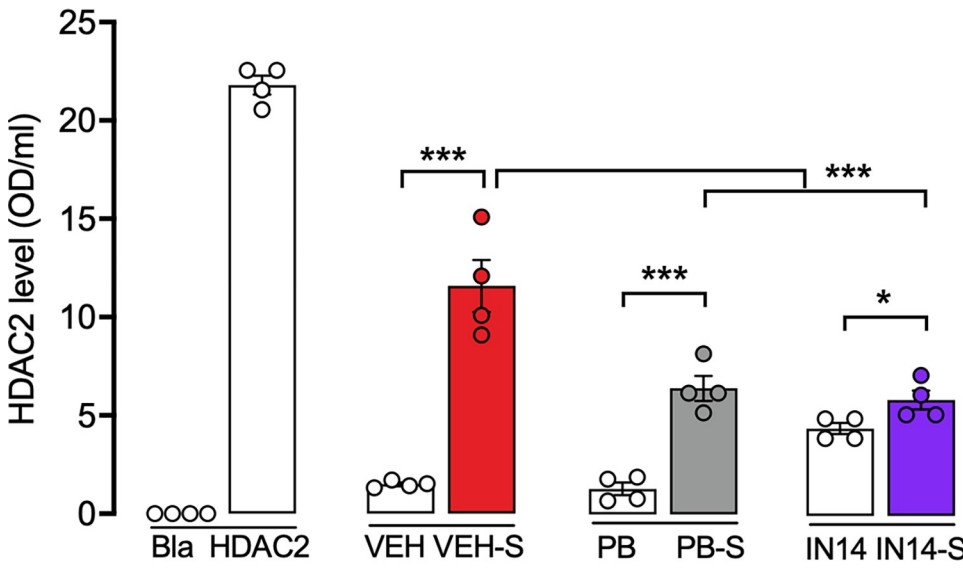

**Fig 8. Increased HDAC2 expression in the hippocampus after swim stress.** IN14 and PB, prevented acute stress-increased expression of HDAC2 in the mouse hippocampus. n = 4. * p < 0.05, *** p< 0.001, Tukey test. Bla: blank; HDAC2: Histone deacetylase-2; VEH: vehicle; PB: 4-phenylbutyrate.

VEH-S (p< 0.01), between VEH-S vs. PB-S (p< 0.05), and VEH-S vs. IN14-S (p = 0.001) groups, where PB-S and IN14-S groups spent more time exploring the target quadrant compared to the VEH-S group. There were no significant differences between the non-stressed groups. Finally, Fig 7D and 7E show the mean (±SEM) of time spent in the target and opposite quadrants during the probe test. One-way analyzes of variance were conducted on the time spent in each of the two quadrants in both non-stressed and stressed animals. One-way ANOVA revealed that there were significant differences in non-stressed and stressed groups (F (5, 54) = 12.33, P<0.0001; F (5, 54) = 13.16, P<0.0001). In the probe test, the duration in the target quadrant of the vehicle (VEH) and the groups treated with PB (-/+ stress), and IN14 (-/+ stress) was remarkably more than that in the opposite quadrant (VEH (T) vs. VEH (O); PB (T) vs. PB (O); IN14 (T) vs. IN14 (O), p< 0.01; PB-S (T) vs. PB(O), p < 0.05, and IN14-S (T) vs. IN14-S (O), p< 0.001, unlike the stressed vehicle (VEH-S), which does not show a significant difference between the quadrants (VEH-S (T) vs. VEH-S (O), ns = 0.1687).

**Acute swim stress increased the hippocampal HDAC2 content.** Fluor de Lys HDAC2 Activity Colorimetric Kit analysis of the hippocampus for total protein levels indicated that the hippocampal HDAC2 protein level was increased in the group subjected to acute stress, showing that the hippocampal HDAC2 level in animals under extrinsic stress was much higher than in the control mice (Fig 8). Two-way ANOVA showed a significant main effect for the drugs treatment (F $_{(2, 9)}$ = 7.208, p = 0.0135), the stress factor (F $_{(1, 9)}$ = 129.4, p = 0.0001) and the drug-stress interaction (F $_{(2, 9)}$ = 26.34, p = 0.0002) on the expression of HDAC2. The post-hoc analysis confirmed that stressed animals had higher HDAC2 expression (VEH vs. VEH-S, t = 7.610, ***p<0.001), and that the administration of PB and IN14 prevented this effect (VEH-S vs. PB-S, p <0.001; VEH-S vs. IN14-S, p = 0.001).

## Discussion

The main findings of the present study were that acute stress induced by forced swimming in mice produced a clear inhibition in memory consolidation in three different learning tasks,

whereas blockade of HDACs induced by two drugs, PB (whose pro-cognitive activity is well documented) and IN14 (a new compound synthesized by our group), were able to reverse the deleterious effect of extrinsic stress on memory. Additionally, forced swimming did induce a notable increase in serum corticosterone levels, which both HDACs inhibitors reversed.

We chose to use three different tasks to assess hippocampus-dependent memory because each explores singular aspects of learning that occur under diverse stress conditions: the ORT is a neutral model (i.e., no emotional load) to assess episodic memory, both short- and long-term, with a single training trial (33), while the ETM allow to assess working memory during multi-trial learning curve construction under anxiogenic conditions, and its long-term retention (37). Finally, the BFLT was adapted by our group to evaluate both short- and long-term spatial memory using an appetitive incentive instead of an aversive one as in other widely used models (40).

The detrimental effect of acute stress 30 min before training on three hippocampus-dependent memory tasks contrasts with its enhancing effect when it is induced during training [41–43] and with the facilitatory effects of intraperitoneal administration of moderate doses of corticosterone immediately post-training [42, 44–46]. Multiple studies in animals and humans have shown that the effect of acute stress on memory critically depends on its temporal relationship with the learning event [7–12]. On one hand, corticosterone facilitates long-term memory consolidation of emotionally arousing experiences, i.e., "corticosterone released within the learning context" [47–49]. In contrast, glucocorticoids negatively influence memory consolidation of information acquired after emotional arousal, i.e., "corticosterone released outside the learning context" [50–52]. Therefore, in our first experiment, it was important to know to what extent acute swimming stress could modify plasma corticosterone levels. As expected, we found that blood corticosterone levels of stressed mice increased significantly 30 minutes after forced swimming, a physiological alteration that has been observed in this test [53, 54]. We also observed that pretreatment with the HDACs inhibitors, IN14 and PB, significantly attenuates the increase of blood corticosterone, same as Vorinostat, an HDAC inhibitor [55].

Epigenetic mechanisms, especially histone acetylation-deacetylation, have been demonstrated to be involved in cognitive regulation and indicate novel regulation patterns associated with stress [56]; considering that IN14 and PB are inhibitors of HDACs [31, 57], it was essential to determine how these compounds are involved in cognitive regulation associated with stress. Consistent with prior reports in the literature [58–60], we found that acute stress-induced before training in three hippocampus-dependent learning tasks impairs long-term memory retention, an effect evidenced by a poor performance in the ORT, ETM and BFLT, 24 or 48 h after training. However, forced swimming stress had only a modest effect on working memory necessary for the acquisition of the ETM and BFLT tasks despite that it affected short-term memory (1 h after training) in the ORT, a similar effect to that shown by Nelissen et al. [52], where acute stress did not affect object recognition memory acquisition but did affect short- and long-term memory. We also found that systemic administration of IN14 and PB, prevented stress-induced memory consolidation disruption, indicating a central role for histone acetylation as an epigenetic mechanism mediating this effect. Histone acetylation makes the transcription of two cognition-related genes, PSD95 and BDNF, substantially easier, which appears to facilitate learning and memory [61, 62]. GRs and HDAC2 are co-localized in hippocampal neurons which suggests that they likely interact with each other [16], while Penney and Tsai [63] identified a glucocorticoid receptor recognition element in the HDAC2 proximal promoter region. Hence, we propose that IN14 and PB reverse the cognitive deficit caused by swim stress through their potential modulatory effect on HDAC2 levels in the hippocampus. However, both drugs also reduced the elevation of blood corticosterone, which

could also contribute to their antianmestic effect. This was a surprising finding that deserves careful analysis because it is crucial to interpreting our results. Research is currently underway to address this limitation of the present study. Yet, it is worth noting that none of the drugs improved memory consolidation in non-stressed animals. This finding may be due to a ceiling effect of vehicle-treated mice performance in both behavioral tasks. However, in a previous study we found that neither IN14 nor PB improved memory in intact mice [31], thus arguing in favor of the view that HDACs inhibitors are not cognitive boosters, but rather potentially useful drugs for mitigating memory deficits in cognitively compromised patients.

The fact that HDACs inhibition prevented the adverse effect of stress on memory consolidation is in agreement with recent studies in rodents showing that systemic administration of Trichostatin A (an HDACs inhibitor) attenuates the detrimental influence of sleep deprivation on memory consolidation [14]. Wu et al. [16] showed that chronic restraint decreased the histone acetylation level in hippocampal neurons, while HDAC2 was augmented and co-localized with glucocorticoid receptors. In models of neurodegenerative diseases, it has been found that pharmacological inhibition of HDACs or HDAC2 silencing prevent histones hypoacetylation and increase gene expression [25, 28, 64], in addition to rescuing or reducing cognitive deficit. The protective effects of IN14 and PB on stress-induced amnesia observed here open the possibility of the potential use of these drugs to prevent the cumulative effects of stressful events on cognitive function.

It has been suggested that the initial release of catecholamines (epinephrine and norepinephrine) induced by acute stress promotes the excitability of neurons in the hippocampus and favors memory consolidation. For instance, noradrenergic activity in the basolateral amygdala enables the facilitation of memory consolidation induced by HDAC inhibition by sodium butyrate in the object recognition and object location memory tasks [65]. When stress induction occurs before training, the noradrenergic neuromodulatory effect on neuronal excitability had already occurred and promoted a refractory phase due to NMDA-type glutamate receptor desensitization and the genomic effects of corticosterone on hippocampal neurons, which precludes memory consolidation [10, 66–68]. These models, however, have not included possible epigenetic mechanisms, such as dynamic changes in histone acetylation in the hippocampus, which may also play a role in these effects [69].

Multiple reports indicate that the increase in GCs levels induces the expression of HDAC2 [1, 14, 15], a protein that can also be associated with GR repressing genes transcription with Glucocorticoid Response Elements (GRE) sequences in their promoter or inducing transrepression of genes without this sequence [70]. In the present study, we explore whether the long-term changes in histone acetylation induced by treatments with HDACs inhibitors IN14 and PB were related to changes in HDAC2 expression in the hippocampus. We found that extrinsic stress increased the long-term expression of HDAC2 in this brain structure and this increase was prevented in mice treated with IN14 and PB. Previous studies have shown that administration of corticosterone in rodents caused a marked increase in the gene expression of HDAC6 and HDAC2 mRNA levels in the hippocampus [55]. Also, exposure to 1.5% isoflurane in pregnant rats rose HDAC2 mRNA levels and decreased CREB mRNA in the hippocampus of the offspring via the HDAC2-CREB-NR2B pathway, impairing the learning and memory of the offspring in the Morris water maze task, effects reversed by vorinostat, a HDAC2 inhibitor [55, 71]. These results, along with ours, confirm that extrinsic stress increases the expression of HDAC2. To further support the role of HDAC2 in learning and long-term memory dysfunction, we showed that treatment with IN14 and PB, before the three behavioral tests explored here, reversed the memory impairment. Although it has been described that the GR/glucocorticoid complex binds to the GRE sequence in the HDAC2 promoter to induce its transcription [1, 15, 24, 63], the epigenetic mechanisms that regulate the

expression of HDAC2 remain unknown. Previous studies have reported that extrinsic stress can reduce the level of histone acetylation and regulate the expression of related target genes [69, 72]. Our results indicate that the induction of HDAC2 expression by extrinsic stress is regulated by histone acetylation and that other epigenetic mechanisms may also be regulating this process. Thus, an increase in HDAC2 expression may subsequently reduce the acetylation of histones in its promoter, reducing its transcription in a homeostatic way.

The present results show that hippocampal HDAC2 expression was higher in stressed mice, which showed poor cognitive performance in ORT, ETM, and BFLT. This finding supports the idea that increased expression of HDAC2 may reduce histone acetylation, leading to decreased transcription of learning- and memory-related genes. Together, our results suggest that HDAC inhibitors could be considered potential therapeutic agents to improve clinical status in cognitively impaired individuals.

## Supporting information

**S1 Data. Original data.**
(XLSX)

## Acknowledgments

We thank, Dr. Jean-Pascal Morin for critical review and edition of the manuscript, Norma Serafín, Martín García, and Ramón García for excellent technical assistance and Bertha Esquivel for administrative support. We also thank Dirección General de Asuntos del Personal Académico de la Universidad Nacional Autónoma de México (DGAPA-UNAM) for Postdoctoral fellowship 2020–2022 to HM-P. We appreciate the valuable comments and suggestions of the anonymous reviewers who greatly contributed to the improvement of our manuscript.

## Author Contributions

**Conceptualization:** Ofir Picazo, Gina L. Quirarte, Gabriel Roldán-Roldán.

**Data curation:** Heidy Martínez-Pacheco, Gabriel Roldán-Roldán.

**Formal analysis:** Heidy Martínez-Pacheco, Gabriel Roldán-Roldán.

**Funding acquisition:** Gina L. Quirarte, Gabriel Roldán-Roldán.

**Investigation:** Heidy Martínez-Pacheco, Rossana Citlali Zepeda, Ofir Picazo, Gina L. Quirarte, Gabriel Roldán-Roldán.

**Methodology:** Heidy Martínez-Pacheco, Ofir Picazo, Gina L. Quirarte, Gabriel Roldán-Roldán.

**Project administration:** Gabriel Roldán-Roldán.

**Resources:** Gina L. Quirarte, Gabriel Roldán-Roldán.

**Software:** Gina L. Quirarte.

**Supervision:** Rossana Citlali Zepeda, Gina L. Quirarte, Gabriel Roldán-Roldán.

**Validation:** Heidy Martínez-Pacheco, Rossana Citlali Zepeda, Gabriel Roldán-Roldán.

**Visualization:** Heidy Martínez-Pacheco, Rossana Citlali Zepeda, Ofir Picazo, Gina L. Quirarte, Gabriel Roldán-Roldán.

**Writing – original draft:** Heidy Martínez-Pacheco, Gina L. Quirarte, Gabriel Roldán-Roldán.

**Writing – review & editing:** Heidy Martínez-Pacheco, Gina L. Quirarte, Gabriel Roldán-Roldán.

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
