## [Decision Letter · Decision Letter 0]

3 Jan 2024

PONE-D-23-35885Class I Histone Deacetylases inhibition reverses memory impairment induced by acute stress in micePLOS ONE

Dear Dr. Quirarte,

Thank you for submitting your manuscript to PLOS ONE. After careful consideration, we feel that it has merit but does not fully meet PLOS ONE’s publication criteria as it currently stands. Therefore, we invite you to submit a revised version of the manuscript that addresses the points raised during the review process.

We look forward to receiving your revised manuscript.

Kind regards,

Peng Zhong, Ph.D.

Academic Editor

PLOS ONE

Journal Requirements:

2.We note that the grant information you provided in the ‘Funding Information’ and ‘Financial Disclosure’ sections do not match. 

"This research was supported by grant DGAPA-PAPIIT-UNAM (IN209822) and a Postdoctoral fellowship from DGAPA-UNAM (2020-2022) to HM-P. "

4. Please expand the acronym “DGAPA-PAPIIT-UNAM” (as indicated in your financial disclosure) so that it states the name of your funders in full.

"We thank, Dr. Jean-Pascal Morin for critical review and edition of the manuscript, Norma Serafín, Martín García, and Ramón García for excellent technical assistance and Bertha Esquivel for administrative support. We also thank DGAPA-UNAM for funding (Grant PAPIIT IN209822, Postdoctoral fellowship 2020-2022 to HM-P), COFAA, and SIP-IPN."

"This research was supported by grant DGAPA-PAPIIT-UNAM (IN209822) and a Postdoctoral fellowship from DGAPA-UNAM (2020-2022) to HM-P. "

6. Thank you for stating the following in your Competing Interests section:  

"The authors declare that they have no known competing financial interests or personal relationships that could have appeared to influence the work reported in this paper."

7. We note that your Data Availability Statement is currently as follows: [All relevant data are within the manuscript and its Supporting Information files.]

8. We note that you have included the phrase “data not shown” in your manuscript. Unfortunately, this does not meet our data sharing requirements. PLOS does not permit references to inaccessible data. We require that authors provide all relevant data within the paper, Supporting Information files, or in an acceptable, public repository. Please add a citation to support this phrase or upload the data that corresponds with these findings to a stable repository (such as Figshare or Dryad) and provide and URLs, DOIs, or accession numbers that may be used to access these data. Or, if the data are not a core part of the research being presented in your study, we ask that you remove the phrase that refers to these data.

Reviewers' comments:

Reviewer's Responses to Questions

**Comments to the Author**

1. Is the manuscript technically sound, and do the data support the conclusions?

Reviewer #1: Yes

Reviewer #2: Partly

2. Has the statistical analysis been performed appropriately and rigorously? 

Reviewer #1: No

Reviewer #2: Yes

3. Have the authors made all data underlying the findings in their manuscript fully available?

Reviewer #1: No

Reviewer #2: Yes

4. Is the manuscript presented in an intelligible fashion and written in standard English?

Reviewer #1: Yes

Reviewer #2: Yes

5. Review Comments to the Author

Reviewer #1: The paper entitled „Class I Histone Deacetylases inhibition reverses memory impairment induced by acute stress in mice” by Martínez-Pacheco et al. demonstrated the possible link between acute stress, memory disturbances and downregulation of HDAC. The topic is quite important due to the complexicity and still not fully understand relationship between the stress and learning and memory processes. The manuscript is well-written but there are some deficiences, which needs to be addressed before the acceptance. All comments are indicated below.

In the Introduction there should be some explanation about the rationale to use HDACs inhibitors in the study (PB and IN14), based on available literature.

In case of animals Author should write the total amount of used animals. Moreover there should be some information about the enrichment used and handling. Authors mentioned about handling only in case of object recognition test, what about the other tests?

Why the spontaneous motor activity was measured for 5 min? Please explain.

„Novel object recognition test” should be actually the „object recognition test”

There are some shortcuts, which are not explained in the text: LTM (line 114), STM (line 126)

Line 167-168 – please explain how this confirmation of animals equall preferences was performed?

Line 183 – Authors write „rats” whereas only mice were included in the Materials and Methods section

„post hoc” should be written in italic

Line 235-236 – please explain why the HDAC2 activity was evaluated onlu in animals from ETM test?

Line 269 p=0.0617 is not significant – please correct the sentence.

Please explain why the number of animals used in case of plasma corticosterone level was only 4 per group?

Line 288-289 – in the Fig.4 in „vertical” – this increase doesn’t indicate to be significant.

Line 320 „mg/kg” with lowercase

Line 339 Authors write Fig.6D, which I didn’t find in the manuscript

In the „Discussion” section there should be more references to performed tests, like short explanation what can be measured in the specific test (please see line 139-141 – these information should be rather in the Discussion, no in the Method section)

Authors should include also the limitations of performed study.

Figure 1 should also include the spontaneous locomotor activity test.

Figure 5C In case of PB-S after 1h – shouldn’t be there any significance vs. VEH-S?

Figure 6B and 7A These charts are poorly legible, please try to correct them.

Figure 7B In case of PB-S– shouldn’t be there any significance vs. VEH-S?

Figure 8B What about the significance between VEH-S and PB-S/IN14-S?

Reviewer #2: The manuscript by Martinez-Pacheco and colleagues studies the effects of acute stress on memory in mice and investigates the role of histone deacetylase 2 (HDAC2) via systemic treatment with HDAC inhibitors. Acute stress was induced by forced swimming, and the increased stress level was confirmed by plasma corticosterone level. Afterwards, three behavioral tests were conducted in independent groups and a negative effect of acute stress on short-term and long-term memory was revealed. Such memory impairment was reversed by HDAC2 inhibitors PB and IN14. In the end, the authors compared the HDAC2 levels in different treatments and found that the memory impairment is paralleled with increased HDAC2 levels. Together, this study claimed that the acute stress induces memory impairment via increased HDAC2 expression, and the memory dysfunction could be reversed by HDAC2 inhibition. The experiments are well designed and can well support their conclusions

---

## [Author Response · Author response to Decision Letter 0]

11 Mar 2024

We have responded to Editor and each of the reviewers' comments and taken into account their valuable suggestions.

---

## [Decision Letter · Decision Letter 1]

3 Apr 2024

Class I Histone Deacetylases inhibition reverses memory impairment induced by acute stress in mice

PONE-D-23-35885R1

Dear Dr. Quirarte,

We’re pleased to inform you that your manuscript has been judged scientifically suitable for publication and will be formally accepted for publication once it meets all outstanding technical requirements.

Kind regards,

Peng Zhong, Ph.D.

Academic Editor

PLOS ONE

Reviewers' comments:

Reviewer's Responses to Questions

**Comments to the Author**

1. If the authors have adequately addressed your comments raised in a previous round of review and you feel that this manuscript is now acceptable for publication, you may indicate that here to bypass the “Comments to the Author” section, enter your conflict of interest statement in the “Confidential to Editor” section, and submit your "Accept" recommendation.

Reviewer #1: All comments have been addressed

Reviewer #2: All comments have been addressed

2. Is the manuscript technically sound, and do the data support the conclusions?

Reviewer #1: (No Response)

Reviewer #2: Yes

3. Has the statistical analysis been performed appropriately and rigorously? 

Reviewer #1: (No Response)

Reviewer #2: Yes

4. Have the authors made all data underlying the findings in their manuscript fully available?

Reviewer #1: (No Response)

Reviewer #2: Yes

5. Is the manuscript presented in an intelligible fashion and written in standard English?

Reviewer #1: (No Response)

Reviewer #2: Yes

6. Review Comments to the Author

Reviewer #1: (No Response)

Reviewer #2: The authors have addressed my concerns in the revised manuscript. This version now is suitable to be published.

7. PLOS authors have the option to publish the peer review history of their article (what does this mean?). If published, this will include your full peer review and any attached files.

Reviewer #1: No

Reviewer #2: No

---

## [Editor Report · Acceptance letter]

8 Apr 2024

PONE-D-23-35885R1 

PLOS ONE

Dear Dr. Quirarte, 

I'm pleased to inform you that your manuscript has been deemed suitable for publication in PLOS ONE. Congratulations! Your manuscript is now being handed over to our production team.

Kind regards, 

on behalf of

Dr. Peng Zhong 

Academic Editor

PLOS ONE